# RANDOM FEATURE ATTENTION

**Hao Peng**[♠][*]  **Nikolaos Pappas**[♠]  **Dani Yogatama**[♣]  **Roy Schwartz**[♡]
**Noah A. Smith**[♠][◇]  **Lingpeng Kong**[♦][*]
[♠]Paul G. Allen School of Computer Science & Engineering, University of Washington
[♣]DeepMind   [◇]Allen Institute for Artificial Intelligence
[♡]School of Computer Science & Engineering, Hebrew University of Jerusalem
[♦]Department of Computer Science , The University of Hong Kong
{hapeng,npappas,nasmith}@cs.washington.edu
dyogatama@google.com, roys@cs.huji.ac.il, lpk@cs.hku.hk

## ABSTRACT

Transformers are state-of-the-art models for a variety of sequence modeling tasks. At their core is an attention function which models pairwise interactions between the inputs at every timestep. While attention is powerful, it does *not* scale efficiently to long sequences due to its quadratic time and space complexity in the sequence length. We propose RFA, a linear time and space **a**ttention that uses **r**andom **f**eature methods to approximate the softmax function, and explore its application in transformers. RFA can be used as a drop-in replacement for conventional softmax attention and offers a straightforward way of learning with recency bias through an optional gating mechanism. Experiments on language modeling and machine translation demonstrate that RFA achieves similar or better performance compared to strong transformer baselines. In the machine translation experiment, RFA decodes twice as fast as a vanilla transformer. Compared to existing efficient transformer variants, RFA is competitive in terms of both accuracy and efficiency on three long text classification datasets. Our analysis shows that RFA's efficiency gains are especially notable on long sequences, suggesting that RFA will be particularly useful in tasks that require working with large inputs, fast decoding speed, or low memory footprints.

## 1 INTRODUCTION

Transformer architectures (Vaswani et al., 2017) have achieved tremendous success on a variety of sequence modeling tasks (Ott et al., 2018; Radford et al., 2018; Parmar et al., 2018; Devlin et al., 2019; Parisotto et al., 2020, *inter alia*). Under the hood, the key component is attention (Bahdanau et al., 2015), which models pairwise interactions of the inputs, regardless of their distances from each other. This comes with quadratic time and memory costs, making the transformers computationally expensive, especially for long sequences. A large body of research has been devoted to improving their time and memory efficiency (Tay et al., 2020c). Although better *asymptotic* complexity and prominent gains for long sequences have been achieved (Lee et al., 2019; Child et al., 2019; Beltagy et al., 2020, *inter alia*), in practice, many existing approaches are less well-suited for moderate-length ones: the additional computation steps required by some approaches can overshadow the time and memory they save (Kitaev et al., 2020; Wang et al., 2020; Roy et al., 2020, *inter alia*).

This work proposes **r**andom **f**eature **a**ttention (RFA), an efficient attention variant that scales linearly in sequence length in terms of time and space, and achieves practical gains for both long and moderate length sequences. RFA builds on a kernel perspective of softmax (Rawat et al., 2019). Using the well-established random feature maps (Rahimi & Recht, 2007; Avron et al., 2016; §2), RFA approximates the dot-then-exponentiate function with a kernel trick (Hofmann et al., 2008): $\exp(\mathbf{x} \cdot \mathbf{y}) \approx \phi(\mathbf{x}) \cdot \phi(\mathbf{y})$. Inspired by its connections to gated recurrent neural networks (Hochreiter & Schmidhuber, 1997; Cho et al., 2014) and fast weights (Schmidhuber, 1992), we further augment RFA with an optional gating mechanism, offering a straightforward way of learning with recency bias when locality is desired.

---

[*]The majority of this work was done while these authors were at DeepMind.

RFA and its gated variant (§3) can be used as a drop-in substitute for the canonical softmax attention, and increase the number of parameters by less than 0.1%. We explore its applications in transformers on language modeling, machine translation, and long text classification (§4). Our experiments show that RFA achieves comparable performance to vanilla transformer baselines in all tasks, while outperforming a recent related approach (Katharopoulos et al., 2020). The gating mechanism proves particularly useful in language modeling: the gated variant of RFA outperforms the transformer baseline on WikiText-103. RFA shines in decoding, even for shorter sequences. In our head-to-head comparison on machine translation benchmarks, RFA decodes around $2\times$ faster than a transformer baseline, *without* accuracy loss. Comparisons to several recent efficient transformer variants on three long text classification datasets show that RFA is competitive in terms of both accuracy and efficiency. Our analysis (§5) shows that more significant time and memory efficiency improvements can be achieved for longer sequences: $12\times$ decoding speedup with less than 10% of the memory for 2,048-length outputs.

## 2 BACKGROUND

### 2.1 ATTENTION IN SEQUENCE MODELING

The attention mechanism (Bahdanau et al., 2015) has been widely used in many sequence modeling tasks. Its dot-product variant is the key building block for the state-of-the-art transformer architectures (Vaswani et al., 2017). Let $\{\mathbf{q}_t\}_{t=1}^N$ denote a sequence of $N$ **query** vectors, that attend to sequences of $M$ **key** and **value** vectors. At each timestep, the attention linearly combines the values weighted by the outputs of a softmax:

$$\mathrm{attn}\left(\mathbf{q}_t, \{\mathbf{k}_i\}, \{\mathbf{v}_i\}\right) = \sum_i \frac{\exp\left(\mathbf{q}_t \cdot \mathbf{k}_i / \tau\right)}{\sum_j \exp\left(\mathbf{q}_t \cdot \mathbf{k}_j / \tau\right)} \mathbf{v}_i^\top. \tag{1}$$

$\tau$ is the temperature hyperparameter determining how "flat" the softmax is (Hinton et al., 2015).[1]

Calculating attention for a single query takes $\mathcal{O}(M)$ time and space. For the full sequence of $N$ queries the space amounts to $\mathcal{O}(MN)$. When the computation *cannot* be parallelized across the queries, e.g., in autoregressive decoding, the time complexity is quadratic in the sequence length.

### 2.2 RANDOM FEATURE METHODS

The theoretical backbone of this work is the unbiased estimation of the Gaussian kernel by Rahimi & Recht (2007). Based on Bochner's theorem (Bochner, 1955), Rahimi & Recht (2007) proposed random Fourier features to approximate a desired shift-invariant kernel. The method nonlinearly transforms a pair of vectors $\mathbf{x}$ and $\mathbf{y}$ using a **random feature map** $\phi$; the inner product between $\phi(\mathbf{x})$ and $\phi(\mathbf{y})$ approximates the kernel evaluation on $\mathbf{x}$ and $\mathbf{y}$. More precisely:

**Theorem 1** (Rahimi & Recht, 2007). *Let $\phi : \mathbb{R}^d \to \mathbb{R}^{2D}$ be a nonlinear transformation:*

$$\phi\left(\mathbf{x}\right) = \sqrt{1/D}\Big[\sin\left(\mathbf{w}_1 \cdot \mathbf{x}\right), \ldots, \sin\left(\mathbf{w}_D \cdot \mathbf{x}\right), \cos\left(\mathbf{w}_1 \cdot \mathbf{x}\right), \ldots, \cos\left(\mathbf{w}_D \cdot \mathbf{x}\right)\Big]^\top. \tag{2}$$

*When $d$-dimensional random vectors $\mathbf{w}_i$ are independently sampled from $\mathcal{N}(\mathbf{0}, \sigma^2 \mathbf{I}_d)$,*

$$\mathbb{E}_{\mathbf{w}_i}\left[\phi\left(\mathbf{x}\right) \cdot \phi\left(\mathbf{y}\right)\right] = \exp\left(-\left\|\mathbf{x} - \mathbf{y}\right\|^2 / 2\sigma^2\right). \tag{3}$$

Variance of the estimation is inversely proportional to $D$ (Appendix A.2; Yu et al., 2016).

Random feature methods proved successful in speeding up kernel methods (Oliva et al., 2015; Avron et al., 2017; Sun, 2019, *inter alia*), and more recently are used to efficiently approximate softmax (Rawat et al., 2019). In §3.1, we use it to derive an unbiased estimate to $\exp(\langle \cdot, \cdot \rangle)$ and then an efficient approximation to softmax attention.

## 3 MODEL

This section presents RFA (§3.1) and its gated variant (§3.2). In §3.3 we lay out several design choices and relate RFA to prior works. We close by practically analyzing RFA's complexity (§3.4).

---

[1] $M = N$ in self-attention; they may differ, e.g., in the cross attention of a sequence-to-sequence model.

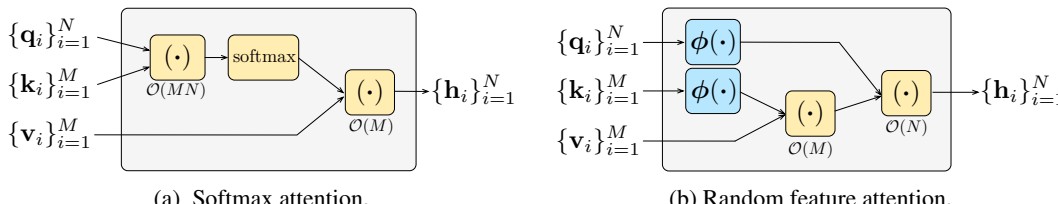

(a) Softmax attention.      (b) Random feature attention.

Figure 1: Computation graphs for softmax attention (left) and random feature attention (right). Here, we assume cross attention with source length $M$ and target length $N$.

## 3.1 RANDOM FEATURE ATTENTION

RFA builds on an unbiased estimate to $\exp(\langle \cdot, \cdot \rangle)$ from Theorem 1, which we begin with:

$$
\begin{aligned}
\exp\left(\mathbf{x} \cdot \mathbf{y}/\sigma^2\right) &= \exp\left(\|\mathbf{x}\|^2 / 2\sigma^2 + \|\mathbf{y}\|^2 / 2\sigma^2\right) \exp\left(-\|\mathbf{x} - \mathbf{y}\|^2 / 2\sigma^2\right) \\
&\approx \exp\left(\|\mathbf{x}\|^2 / 2\sigma^2 + \|\mathbf{y}\|^2 / 2\sigma^2\right) \boldsymbol{\phi}\left(\mathbf{x}\right) \cdot \boldsymbol{\phi}\left(\mathbf{y}\right).
\end{aligned}
\tag{4}
$$

The last line does *not* have any nonlinear interaction between $\boldsymbol{\phi}(\mathbf{x})$ and $\boldsymbol{\phi}(\mathbf{y})$, allowing for a linear time/space approximation to attention. For clarity we assume the query and keys are unit vectors.[2]

$$
\begin{aligned}
\operatorname{attn}\left(\mathbf{q}_t, \{\mathbf{k}_i\}, \{\mathbf{v}_i\}\right) &= \sum_i \frac{\exp\left(\mathbf{q}_t \cdot \mathbf{k}_i/\sigma^2\right)}{\sum_j \exp\left(\mathbf{q}_t \cdot \mathbf{k}_j/\sigma^2\right)} \mathbf{v}_i^\top \\
&\approx \sum_i \frac{\boldsymbol{\phi}\left(\mathbf{q}_t\right)^\top \boldsymbol{\phi}\left(\mathbf{k}_i\right) \mathbf{v}_i^\top}{\sum_j \boldsymbol{\phi}\left(\mathbf{q}_t\right) \cdot \boldsymbol{\phi}\left(\mathbf{k}_j\right)} \\
&= \frac{\boldsymbol{\phi}\left(\mathbf{q}_t\right)^\top \sum_i \boldsymbol{\phi}\left(\mathbf{k}_i\right) \otimes \mathbf{v}_i}{\boldsymbol{\phi}\left(\mathbf{q}_t\right) \cdot \sum_j \boldsymbol{\phi}\left(\mathbf{k}_j\right)} = \operatorname{RFA}\left(\mathbf{q}_t, \{\mathbf{k}_i\}, \{\mathbf{v}_i\}\right).
\end{aligned}
\tag{5}
$$

$\otimes$ denotes the outer product between vectors, and $\sigma^2$ corresponds to the temperature term $\tau$ in Eq. 1.

RFA can be used as a drop-in-replacement for softmax-attention.

(a) The input is revealed in full to **cross attention** and **encoder self-attention**. Here RFA calculates attention using Eq. 5.
(b) In **causal attention** RFA attends only to the prefix.[3] This allows for a recurrent computation. Tuple $(\mathbf{S}_t \in \mathbb{R}^{2D \times d}, \mathbf{z}_t \in \mathbb{R}^{2D})$ is used as the "hidden state" at time step $t$ to keep track of the history, similar to those in RNNs. Then $\operatorname{RFA}(\mathbf{q}_t, \{\mathbf{k}_i\}_{i \leq t}, \{\mathbf{v}_i\}_{i \leq t}) = \boldsymbol{\phi}(\mathbf{q}_t)^\top \mathbf{S}_t / (\boldsymbol{\phi}(\mathbf{q}_t) \cdot \mathbf{z}_t)$, where

$$
\mathbf{S}_t = \mathbf{S}_{t-1} + \boldsymbol{\phi}\left(\mathbf{k}_t\right) \otimes \mathbf{v}_t, \quad \mathbf{z}_t = \mathbf{z}_{t-1} + \boldsymbol{\phi}\left(\mathbf{k}_t\right).
\tag{6}
$$

$2D$ denotes the size of $\boldsymbol{\phi}(\cdot)$. Appendix A.1 summarizes the computation procedure of RFA, and Figure 1 compares it against the softmax attention. Appendix A.3 derives causal RFA in detail.

Analogously to the softmax attention, RFA has its multiheaded variant (Vaswani et al., 2017). In our experiments we use causal RFA in a transformer language model (§4.1), and both cross and causal RFA in the decoder of a sequence-to-sequence machine translation model.

## 3.2 RFA-GATE: LEARNING WITH RECENCY BIAS

The canonical softmax attention does *not* have any explicit modeling of distance or locality. In learning problems where such inductive bias is crucial (Ba et al., 2016; Parmar et al., 2018; Miconi et al., 2018; Li et al., 2019, *inter alia*), transformers heavily rely on positional encodings. Answering to this, many approaches have been proposed, e.g., learning the attention spans (Sukhbaatar et al.,

---

[2]This can be achieved by $\ell_2$-normalizing the query and keys. See §3.3 for a related discussion.
[3]It is also sometimes called "decoder self-attention" or "autoregressive attention."

2019; Wu et al., 2020), and enhancing the attention computation with recurrent (Hao et al., 2019; Chen et al., 2019) or convolutional (Wu et al., 2019; Mohamed et al., 2019) components.

RFA faces the same issue, but its causal attention variant (Eq. 6) offers a straightforward way of learning with recency bias. We draw inspiration from its connections to RNNs, and augment RFA with a learned gating mechanism (Hochreiter & Schmidhuber, 1997; Cho et al., 2014; Peng et al., 2018, *inter alia*):

$$
\begin{aligned}
g_t &= \text{sigmoid}(\mathbf{w}_g \cdot \mathbf{x}_t + b_g), \\
\mathbf{S}_t &= g_t \, \mathbf{S}_{t-1} + (1 - g_t) \, \boldsymbol{\phi}\left(\mathbf{k}_t\right) \otimes \mathbf{v}_t, \\
\mathbf{z}_t &= g_t \, \mathbf{z}_{t-1} + (1 - g_t) \, \boldsymbol{\phi}\left(\mathbf{k}_t\right).
\end{aligned}
\tag{7}
$$

$\mathbf{w}_g$ and $b_g$ are learned parameters, and $\mathbf{x}_t$ is the input representation at timestep $t$.[4] By multiplying the learned scalar gates $0 < g_t < 1$ against the hidden state $(\mathbf{S}_t, \mathbf{z}_t)$, history is exponentially decayed, favoring more recent context.

The gating mechanism shows another benefit of RFA: it would be otherwise more difficult to build similar techniques into the softmax attention, where there is no clear sense of "recurrence" (Appendix A.5). It proves useful in our language modeling experiments (§4.1).

### 3.3 DISCUSSION

**On query and key norms, and learned random feature variance.** Eq. 5 assumes both the query and keys are of norm-1. It therefore approximates a softmax attention that normalizes the queries and keys before multiplying them, and then scales the logits by dividing them by $\sigma^2$. Empirically, this normalization step scales down the logits (Vaswani et al., 2017) and enforces that $-1 \le \mathbf{q}^\top \mathbf{k} \le 1$. In consequence, the softmax outputs would be "flattened" if not for $\sigma$, which can be set *a priori* as a hyperparameter (Yu et al., 2016; Avron et al., 2017; Sun, 2019, *inter alia*). Here we instead learn it from data with the reparameterization trick (Kingma & Welling, 2014):

$$
\widetilde{\mathbf{w}}_i \sim \mathcal{N}(\mathbf{0}, \mathbf{I}_d), \quad \mathbf{w}_i = \boldsymbol{\sigma} \circ \widetilde{\mathbf{w}}_i.
\tag{8}
$$

$\mathbf{I}_d$ is the $d \times d$ identity matrix, and $\circ$ denotes elementwise product between vectors. $d$-dimensional vector $\boldsymbol{\sigma}$ is learned, but random vectors $\widetilde{\mathbf{w}}_i$ are *not*.[5]

This norm-1 constraint is never mandatory. Rather, we employ it for notation clarity and easier implementation. In preliminary experiments we find it has little impact on the performance when $\sigma$ is set properly or learned from data. Eq. 12 in Appendix A presents RFA *without* imposing it.

**Going beyond the Gaussian kernel.** More broadly, random feature methods can be applied to a family of shift-invariant kernels, with the Gaussian kernel being one of them. In the same family, the order-1 arc-cosine kernel (Cho & Saul, 2009) can be approximated with feature map: $\phi_{\text{arccos}}(\mathbf{x}) = \sqrt{1/D}[\text{ReLU}(\mathbf{w}_1 \cdot \mathbf{x}), \dots, \text{ReLU}(\mathbf{w}_D \cdot \mathbf{x})]^\top$ (Alber et al., 2017).[6] In our experiments, the Gaussian and arc-cosine variants achieve similar performance. This supplements the exploration of alternatives to softmax in attention (Tsai et al., 2019; Gao et al., 2019).

**Relations to prior work.** Katharopoulos et al. (2020) inspire the causal attention variant of RFA. They use a feature map based on the exponential linear unit activation (Clevert et al., 2016): $\text{elu}(\cdot) + 1$. It significantly *underperforms* both the baseline and RFA in our controlled experiments, showing the importance of a properly-chosen feature map. Random feature approximation of attention is also explored by a concurrent work (Choromanski et al., 2021), with applications in masked language modeling for proteins. They propose positive random features to approximate softmax, aiming for a lower variance in critical regions. RFA instead normalizes the queries and keys before random projection to reduce variance. Going beyond both, RFA establishes the benefits of random feature methods as a more universal substitute for softmax across all attention variants, facilitating its applications in, e.g., sequence-to-sequence learning.

---

[4]In multihead attention (Vaswani et al., 2017), $\mathbf{k}_t$ and $\mathbf{v}_t$ are calculated from $\mathbf{x}_t$ using learned affine transformations.

[5]This departs from Eq. 2 by lifting the isotropic assumption imposed on the Gaussian distribution: note the difference between the vector $\boldsymbol{\sigma}$ in Eq. 8 and the scalar $\sigma$ in Eq. 3. We find this improves the performance in practice (§4), even though the same result in Theorem 1 may not directly apply.

[6]Apart from replacing the sinusoid functions with $\text{ReLU}$, it constructs $\mathbf{w}_i$ in the same way as Eq. 8.

There are interesting connections between gated RFA and fast weights (Schmidhuber, 1992; 1993; Ba et al., 2016; Miconi et al., 2018, *inter alia*). Emphasizing recent patterns, they learn a temporal memory to store history similarly to Eqs. 7. The main difference is that RFA additionally normalizes the output using $\phi(\mathbf{q}_t) \cdot \mathbf{z}$ as in Eq. 6, a by-product of approximating softmax's partition function. It is intriguing to study the role of this normalization term, which we leave to future work.

### 3.4 COMPLEXITY ANALYSIS

**Time.** Scaling linearly in the sequence lengths, RFA needs less computation (in terms of number of operations) for long sequences. This implies speedup wherever the quadratic-time softmax attention *cannot* be fully-parallelized across time steps. More specifically:

- Significant speedup can be expected in autoregressive *decoding*, both conditional (e.g., machine translation) and unconditional (e.g., sampling from a language model). For example, $1.9\times$ speedup is achieved in our machine translation experiments (§4.2); and more for longer sequences (e.g., $12\times$ for 2,048-length ones; §5).
- Some applications (e.g., language modeling, text classification) reveal inputs to the model in full.[7] When there are enough threads to parallelize softmax attention across time steps, hardly any speedup from RFA can be achieved; when there are not, typically for very long sequences ($>1,000$), substantial speed gain is possible. For example, RFA does *not* achieve any speedup when working with 512-length context (§4.1), but achieves a $5.3\times$ speedup with 4,000-length context (§4.2).

**Memory.** Asymptotically, RFA has a better memory efficiency than its softmax counterpart (linear vs. quadratic). To reach a more practical conclusion, we include in our analysis the cost of the feature maps. $\phi$'s memory overhead largely depends on its size $D$. For example, let's consider the cross attention of a decoder. RFA uses $\mathcal{O}(4D + 2Dd)$ space to store $\phi(\mathbf{q}_t)$, $\sum_i \phi(\mathbf{k}_i) \otimes \mathbf{v}_i$, and $\sum_i \phi(\mathbf{k}_i)$ (Eq. 5; line 12 of Algo. 2).[8] In contrast, softmax cross attention stores the encoder outputs with $\mathcal{O}(Md)$ memory, with $M$ being the source length. In this case RFA has a lower memory overhead when $2D \ll M$. Typically $D$ should be no less than $d$ in order for reasonable approximation (Yu et al., 2016); In a transformer model, $d$ is the size of an attention head, which is usually around 64 or 128 (Vaswani et al., 2017; Ott et al., 2018). This suggests that RFA can achieve significant memory saving with longer sequences, which is supported by our empirical analysis in §5. Further, using moderate sized feature maps is also desirable, so that its overhead does not overshadow the time and memory RFA saves. We experiment with $D$ at $d$ and $2d$; the benefit of using $D > 2d$ is marginal.

Appendix A.6 discusses the time and space complexity in more detail, and Appendix C.2 studies the effect of random feature size on performance.

## 4 EXPERIMENTS

We evaluate RFA on language modeling, machine translation, and long text classification.

### 4.1 LANGUAGE MODELING

**Setting.** We experiment with WikiText-103 (Merity et al., 2017). It is based on English Wikipedia. Table 5 in Appendix B summarizes some of its statistics. We compare the following models:

- BASE is our implementation of the strong transformer-based language model by Baevski & Auli (2019).
- RFA builds on BASE, but replaces the softmax attention with random feature attention. We experiment with both Gaussian and arc-cosine kernel variants.
- RFA-GATE additionally learns a sigmoid gate on top of RFA (§3.2). It also has a Gaussian kernel variant and a arc-cosine kernel one.[9]
- $\phi_{\mathrm{elu}}$ is a baseline to RFA. Instead of the random feature methods it uses the $\mathrm{elu}(\cdot) + 1$ feature map, as in Katharopoulos et al. (2020).

---

[7]A causal masking is usually used to prevent the model from accessing future tokens in language models.

[8]RFA *never* constructs the $M \times 2D \times d$ tensor $[\phi(\mathbf{k}_i) \otimes \mathbf{v}_i]_i$, but sequentially processes the sequence.

[9]This gating technique is specific to RFA variants, in the sense that it is less intuitive to apply it in BASE.

To ensure fair comparisons, we use comparable implementations, tuning, and training procedure. All models use a 512 block size during both training and evaluation, i.e., they read as input a segment of 512 consecutive tokens, *without* access to the context from previous mini-batches. RFA variants use 64-dimensional random feature maps. We experiment with two model size settings, **small** (around 38M parameters) and **big** (around 242M parameters); they are described in Appendix B.1 along with other implementation details.

| Model | Small | | Big | |
|---|---|---|---|---|
| | Dev. | Test | Dev. | Test |
| BASE | 33.0 | 34.5 | 24.5 | 26.2 |
| $\phi_{\text{elu}}$ (Katharopoulos et al., 2020) | 38.4 | 40.1 | 28.7 | 30.2 |
| RFA-Gaussian | 33.6 | 35.7 | 25.8 | 27.5 |
| RFA-arccos | 36.0 | 37.7 | 26.4 | 28.1 |
| RFA-GATE-Gaussian | **31.3** | **32.7** | **23.2** | **25.0** |
| RFA-GATE-arccos | **32.8** | **34.0** | 24.8 | 26.3 |
| RFA-GATE-Gaussian-Stateful | **29.4** | **30.5** | **22.0** | **23.5** |

Table 1: Language model perplexity (lower is better) on the WikiText-103 development and test sets. Bolded numbers outperform BASE.

**Results.** Table 1 compares the models' performance in perplexity on WikiText-103 development and test data. Both kernel variants of RFA, *without* gating, outperform $\phi_{\text{elu}}$ by more than 2.4 and 2.1 test perplexity for the small and big model respectively, confirming the benefits from using random feature approximation.[10] Yet both *underperform* BASE, with RFA-Gaussian having a smaller gap. Comparing RFA against its gated variants, a more than 1.8 perplexity improvement can be attributed to the gating mechanism; and the gap is larger for small models. Notably, RFA-GATE-Gaussian outperforms BASE under both size settings by at least 1.2 perplexity. In general, RFA models with Gaussian feature maps outperform their arc-cosine counterparts.[11] From the analysis in §3.4 we would *not* expect speedup by RFA models, nor do we see any in the experiments.[12]

Closing this section, we explore a "stateful" variant of RFA-GATE-Gaussian. It passes the last hidden state $(\mathbf{S}_t, \mathbf{z}_t)$ to the next mini-batch during both training and evaluation, a technique commonly used in RNN language models (Merity et al., 2018). This is a consequence of RFA's RNN-style computation, and is less straightforward to be applicable in the vanilla transformer models.[13] From the last row of Table 1 we see that this brings a more than 1.5 test perplexity improvement.

## 4.2 MACHINE TRANSLATION

**Datasets.** We experiment with three standard machine translation datasets.

- WMT14 EN-DE and EN-FR (Bojar et al., 2014). Our data split and preprocessing follow those of Vaswani et al. (2017). We share the source and target vocabularies within each language pair, with 32,768 byte pair encoding types (BPE; Sennrich et al., 2016).
- IWSLT14 DE-EN (Cettolo et al., 2014) is based on TED talks. The preprocessing follows Edunov et al. (2018). Separate vocabularies of 9K/7K BPE types are used for the source and target.

Table 5 in Appendix B summarizes some statistics of the datasets.

---

[10]All models are trained for 150K steps; this could be part of the reason behind the suboptimal performance of $\phi_{\text{elu}}$: it may need 3 times more gradient updates to reach similar performance to the softmax attention baseline (Katharopoulos et al., 2020).

[11]We observe that RFA Gaussian variants are more stable and easier to train than the arc-cosine ones as well as $\phi_{\text{elu}}$. We conjecture that this is because the outputs of the Gaussian feature maps have an $\ell_2$-norm of 1, which can help stabilize training. To see why, $\sin^2(x) + \cos^2(x) = \cos(x - x) = 1$.

[12]In fact, RFA *trains* around 15% slower than BASE due to the additional overhead from the feature maps.

[13]Some transformer models use a text segment from the previous mini-batch as a prefix (Baevski & Auli, 2019; Dai et al., 2019). Unlike RFA, this gives the model access to only a limited amount of context, and significantly increases the memory overhead.

**Setting.** We compare the RFA variants described in §4.1. They build on a BASE model that is our implementation of the base-sized transformer (Vaswani et al., 2017). All RFA models apply random feature attention in decoder cross and causal attention, but use softmax attention in encoders. This setting yields the greatest decoding time and memory savings (§3.4). We use 128/64 for $D$ in cross/causal attention. RFA-GATE learns sigmoid gates in the decoder causal attention. The $\phi_{\text{elu}}$ baseline uses the same setting and applies feature map in both decoder cross and causal attention, but *not* in the encoders. Further details are described in Appendix B.2.

| Model | WMT14 | | IWSLT14 | |
| | EN-DE | EN-FR | DE-EN | Speed |
|---|---|---|---|---|
| BASE | 28.1 | 39.0 | 34.6 | 1.0× |
| $\phi_{\text{elu}}$ (Katharopoulos et al., 2020) | 21.3 | 34.0 | 29.9 | 2.0× |
| RFA-Gaussian | 28.0 | 39.2 | 34.5 | 1.8× |
| RFA-arccos | 28.1 | 38.9 | 34.4 | 1.9× |
| RFA-GATE-Gaussian | 28.1 | 39.0 | 34.6 | 1.8× |
| RFA-GATE-arccos | 28.2 | 39.2 | 34.4 | 1.9× |

Table 2: Machine translation test set BLEU. The decoding speed (last column) is relative to BASE. All models are tested on a single TPU v2 accelerator, with batch size 32.

**Results.** Table 2 compares the models' test set BLEU on three machine translation datasets. Overall both Gaussian and arc-cosine variants of RFA achieve similar performance to BASE on all three datasets, significantly outperforming Katharopoulos et al. (2020). Differently from the trends in the language modeling experiments, here the gating mechanism does not lead to substantial gains. Notably, all RFA variants decode more than $1.8\times$ faster than BASE.

## 4.3 LONG TEXT CLASSIFICATION

We further evaluate RFA's accuracy and efficiency when used as text encoders on three NLP tasks from the recently proposed Long Range Arena benchmark (Tay et al., 2021), designed to evaluate efficient Transformer variants on tasks that require processing long sequences.[14]

**Experimental setting and datasets.** We compare RFA against baselines on the following datasets:

- ListOps (**LO**; Nangia & Bowman, 2018) aims to diagnose the capability of modelling hierarchically structured data. Given a sequence of operations on single-digit integers, the model predicts the solution, also a single-digit integer. It is formulated as a 10-way classification. We follow Tay et al. (2021) and consider sequences with 500–2,000 symbols.
- Character-level text classification with the **IMDb** movie review dataset (Maas et al., 2011). This is a binary sentiment classification task.
- Character-level document retrieval with the ACL Anthology Network (**AAN**; Radev et al., 2009) dataset. The model classifies whether there is a citation between a pair of papers.

To ensure fair comparisons, we implement RFA on top of the transformer baseline by Tay et al. (2021), and closely follow their preprocessing, data split, model size, and training procedure. Speed and memory are evaluated on the IMDb dataset. For our RFA model, we use $D = 64$ for the IMDb dataset, and $D = 128$ for others. We refer the readers to Tay et al. (2021) for further details.

**Results.** From Table 3 we can see that RFA outperforms the transformer baseline on two out of the three datasets, achieving the best performance on IMDb with 66% accuracy. Averaging across three datasets, RFA outperforms the transformer by 0.3% accuracy, second only to Zaheer et al. (2020) with a 0.1% accuracy gap. In terms of time and memory efficiency, RFA is among the strongest. RFA speeds up over the transformer by $1.1$–$5.3\times$, varying by sequence length. Importantly, compared to the only two baselines that perform comparably to the baseline transformer model (Tay et al., 2020a; Zaheer et al., 2020), RFA has a clear advantage in both speed and memory efficiency, and is the only model that is competitive in both accuracy and efficiency.

---

[14]https://github.com/google-research/long-range-arena

| Model | Accuracy | | | | Speed | | | | Memory | | | |
|---|---|---|---|---|---|---|---|---|---|---|---|---|
| | LO | IMDb | AAN | Avg. | 1K | 2K | 3K | 4K | 1K | 2K | 3K | 4K |
| Transformer | 36.4 | 64.3 | 57.5 | 52.7 | 1.0 | 1.0 | 1.0 | 1.0 | 1.00 | 1.00 | 1.00 | 1.00 |
| Wang et al. (2020) | 35.7 | 53.9 | 52.3 | 47.3 | 1.2 | 1.9 | 3.7 | 5.5 | **0.44** | **0.21** | 0.18 | **0.10** |
| Kitaev et al. (2020) | **37.3** | 56.1 | 53.4 | 48.9 | 0.5 | 0.4 | 0.7 | 0.8 | 0.56 | 0.37 | 0.28 | 0.24 |
| Tay et al. (2020b) | 17.1 | 63.6 | **59.6** | 46.8 | 1.1 | 1.6 | 2.9 | 3.8 | 0.55 | 0.31 | 0.20 | 0.16 |
| Tay et al. (2020a) | 37.0 | 61.7 | 54.7 | 51.1 | 1.1 | 1.2 | 2.9 | 1.4 | 0.76 | 0.75 | 0.74 | 0.74 |
| Zaheer et al. (2020) | 36.0 | 64.0 | 59.3 | **53.1** | 0.9 | 0.8 | 1.2 | 1.1 | 0.90 | 0.56 | 0.40 | 0.30 |
| Katharopoulos et al. (2020) | 16.1 | 65.9 | 53.1 | 45.0 | 1.1 | **1.9** | 3.7 | 5.6 | 0.44 | 0.22 | **0.14** | 0.11 |
| Choromanski et al. (2021) | 18.0 | 65.4 | 53.8 | 45.7 | **1.2** | **1.9** | **3.8** | **5.7** | 0.44 | 0.22 | 0.15 | 0.11 |
| RFA-Gaussian (This work) | 36.8 | **66.0** | 56.1 | 53.0 | 1.1 | 1.7 | 3.4 | 5.3 | 0.53 | 0.30 | 0.21 | 0.16 |

Table 3: Accuracy (higher is better) of different models on LO, IMDb, and AAN, along with their speed (higher is better) and peak memory consumption (lower is better) varying sequence lengths (1–4K). Speed and memory are evaluated on the IMDb dataset and relative to the transformer's. Bold font indicates the best performance in each column, and underlined numbers outperform the transformer in accuracy. Transformer's and previous works' numbers are due to Tay et al. (2021).

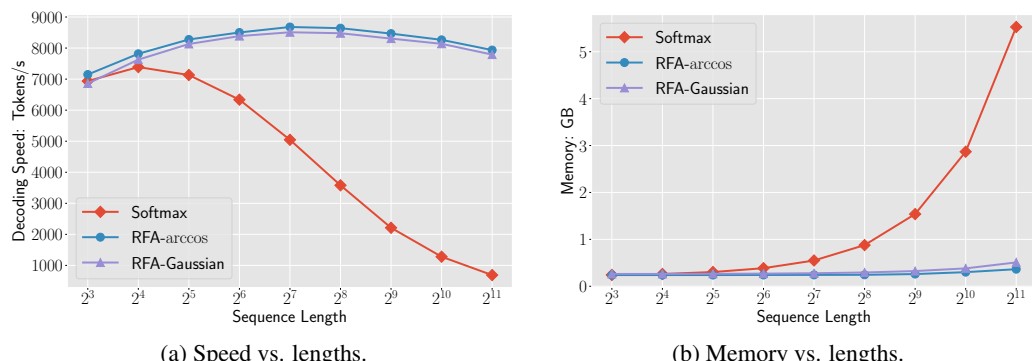

(a) Speed vs. lengths.         (b) Memory vs. lengths.

Figure 2: Conditional decoding speed (left) and memory overhead (right) varying the output lengths. All models are tested on a single TPU v2 accelerator, with greedy decoding and batch size 16.

## 5 ANALYSIS

**Decoding time and memory varying by sequence length.** §3.4 shows that RFA can potentially achieve more significant speedup and memory saving for longer sequences, which we now explore.

We use a simulation conditional generation experiment on to compare RFA's sequence-to-sequence decoding speed and memory overhead against the baseline's. Here we assume the input and output sequences are of the same length. The compared models are of the same size as those described in §4.2, with 6-layer encoders and decoders. Other hyperparameters are summarized in Appendix B.2. All models are tested using greedy decoding with the same batch size of 16, on a TPU v2 accelerator.

From Figures 2 (a) and (b) we observe clear trends. Varying the lengths, both RFA variants achieve consistent decoding speed with nearly-constant memory overhead. In contrast, the baseline decodes slower for longer sequences, taking an increasing amount of memory. Notably, for 2,048-length sequences, RFA decodes around $12\times$ faster than the baseline while using less than 10% of the memory. RFA-arccos slightly outperforms RFA-Gaussian in terms of speed and memory efficiency. This is because when using the same $D$ (as we do here), the $\phi_{\text{arccos}}$ is half the size of $\phi_{\text{Gaussian}}$. These results suggest that RFA can be particularly useful in sequence-to-sequence tasks with longer sequences, e.g., document-level machine translation (Miculicich et al., 2018).

Figure 3 in Appendix C.1 compares the speed and memory consumption in *unconditional* decoding (e.g., sampling from a language model). The overall trends are similar to those in Figure 2.

**Notes on decoding speed.** With a lower memory overhead, RFA can use a larger batch size than the baseline. As noted by Katharopoulos et al. (2020) and Kasai et al. (2021), if we had used mini-

batches as large as the hardware allows, RFA could have achieved a more significant speed gain. Nonetheless, we control for batch size even though it is not the most favorable setting for RFA, since the conclusion translates better to common applications where one generates a single sequence at a time (e.g., instantaneous machine translation). For the softmax attention baseline, we follow Ott et al. (2018) and cache previously computed query/key/value representations, which significantly improves its decoding speed (over not caching).

**Further analysis results.** RFA achieves comparable performance to softmax attention. Appendix C.3 empirically shows that this *cannot* be attributed to RFA learning a good approximation to softmax: when we train with one attention but evaluate with the other, the performance is hardly better than randomly-initialized untrained models. Yet, an RFA model initialized from a pretrained softmax transformer achieves decent training loss after a moderate amount of finetuning steps (Appendix C.4). This suggests some potential applications, e.g., transferring knowledge from a pretrained transformer (e.g., GPT-3; Brown et al., 2020) to an RFA model that is more efficient to sample from.

# 6    RELATED WORK

One common motivation across the following studies, that is shared by this work and the research we have already discussed, is to scale transformers to long sequences. Note that there are plenty orthogonal choices for improving efficiency such as weight sharing (Dehghani et al., 2019), quantization (Shen et al., 2020), knowledge distillation (Sanh et al., 2020), and adapters (Houlsby et al., 2019). For a detailed overview we refer the reader to Tay et al. (2020c).

**Sparse attention patterns.** The idea behind these methods is to limit the reception field of attention computation. It motivates earlier attempts in improving attention's efficiency, and still receives lots of interest. The sparse patterns can be set *a priori* (Liu et al., 2018; Qiu et al., 2020; Ho et al., 2020; You et al., 2020, *inter alia*) or learned from data (Sukhbaatar et al., 2019; Roy et al., 2020, *inter alia*). For most of these approaches, it is yet to be empirically verified that they are suitable for large-scale sequence-to-sequence learning; few of them have recorded decoding speed benefits.

**Compressed context.** Wang et al. (2020) compress the context along the timesteps so that the effective sequence length for attention computation is reduced. Another line of work aims to store past context into a memory module with limited size (Lee et al., 2019; Ainslie et al., 2020; Rae et al., 2020, *inter alia*), so that accessing longer history only moderately increases the overhead. Reminiscent of RNN language models, RFA attends beyond a fixed context window through a stateful computation, *without* increasing time or memory overhead.

# 7    CONCLUSION

We presented random feature attention (RFA). It views the softmax attention through the lens of kernel methods, and approximates it with random feature methods. With an optional gating mechanism, RFA provides a straightforward way of learning with recency bias. RFA's time and space complexity is linear in the sequence length. We use RFA as a drop-in substitute for softmax attention in transformer models. On language modeling, machine translation, and long text classification benchmarks, RFA achieves comparable or better performance than strong baselines. In the machine translation experiment, RFA decodes twice as fast. Further time and memory efficiency improvements can be achieved for longer sequences.

## ACKNOWLEDGMENTS

We would like to thank Phil Blunsom, Chris Dyer, Nando de Freitas, Jungo Kasai, Adhiguna Kuncoro, Dianqi Li, Ofir Press, Lianhui Qin, Swabha Swayamdipta, Sam Thomson, the language team at DeepMind and the ARK group at the University of Washington for their helpful feedback. We also thank Tay Yi for helping run the Long Range Arena experiments, Richard Tanburn for the advice on implementations, and the anonymous reviewers for their thoughtful comments. This work was supported in part by NSF grant 1562364 and a Google Fellowship. Nikolaos Pappas was supported by the Swiss National Science Foundation under grant number P400P2_183911 "UNISON."

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

# Appendices

## A  RANDOM FEATURE ATTENTION IN MORE DETAIL

### A.1  DETAILED COMPUTATION PROCEDURE

Algorithms 1 and 2 describe causal and cross random feature attention's computation procedures.

---

**Algorithm 1** Causal random feature attention.

1: **procedure** RFA-CAUSAL( $\{\mathbf{q}_i\}_{i=1}^N$, $\{\mathbf{k}_i\}_{i=1}^N$, $\{\mathbf{v}_i\}_{i=1}^N$ )
2:    ▷ $\mathbf{S}$ is a $D \times d$ matrix
3:    ▷ $\mathbf{z}$ is a $D$-dimensional vector
4:    $\mathbf{S}, \mathbf{z} \leftarrow \mathbf{0}, \mathbf{0}$
5:    **for** $i = 1$ **to** $N$ **do**
6:       $\widetilde{\mathbf{q}}_i, \widetilde{\mathbf{k}}_i \leftarrow \boldsymbol{\phi}(\mathbf{q}_i), \boldsymbol{\phi}(\mathbf{k}_i)$    ▷ Random feature maps
7:       $\mathbf{S} \leftarrow \mathbf{S} + \widetilde{\mathbf{k}}_i \otimes \mathbf{v}_i$
8:       $\mathbf{z} \leftarrow \mathbf{z} + \widetilde{\mathbf{k}}_i$
9:       $\mathbf{h}_i^\top \leftarrow \widetilde{\mathbf{q}}_i^\top \mathbf{S} / (\widetilde{\mathbf{q}}_i \cdot \mathbf{z})$
10:   **end for**
11:   **return** $\{\mathbf{h}_i\}_{i=1}^N$
12: **end procedure**

---

**Algorithm 2** Cross random feature attention.

1: **procedure** RFA-CROSS( $\{\mathbf{q}_i\}_{i=1}^N$, $\{\mathbf{k}_i\}_{i=1}^M$, $\{\mathbf{v}_i\}_{i=1}^M$ )
2:    ▷ $\mathbf{S}$ is a $D \times d$ matrix
3:    ▷ $\mathbf{z}$ is a $D$-dimensional vector
4:    $\mathbf{S}, \mathbf{z} \leftarrow \mathbf{0}, \mathbf{0}$
5:    **for** $i = 1$ **to** $M$ **do**
6:       $\widetilde{\mathbf{k}}_i \leftarrow \boldsymbol{\phi}(\mathbf{k}_i)$    ▷ Random feature map
7:       $\mathbf{S} \leftarrow \mathbf{S} + \widetilde{\mathbf{k}}_i \otimes \mathbf{v}_i^\top$
8:       $\mathbf{z} \leftarrow \mathbf{z} + \widetilde{\mathbf{k}}_i$
9:    **end for**
10:   **for** $i = 1$ **to** $N$ **do**
11:      $\widetilde{\mathbf{q}}_i \leftarrow \boldsymbol{\phi}(\mathbf{q}_i)$    ▷ Random feature map
12:      $\mathbf{h}_i^\top \leftarrow \widetilde{\mathbf{q}}_i^\top \mathbf{S} / (\widetilde{\mathbf{q}}_i \cdot \mathbf{z})$
13:   **end for**
14:   **return** $\{\mathbf{h}_i\}_{i=1}^N$
15: **end procedure**

---

### A.2  VARIANCE OF RANDOM FOURIER FEATURES

The following result is due to Yu et al. (2016). Using the same notation as in §2.2:

$$\text{Var}(\boldsymbol{\phi}(\mathbf{x}) \cdot \boldsymbol{\phi}(\mathbf{y})) = \frac{1}{2D} \left(1 - e^{-z^2}\right)^2,$$  (9)

where $z = \|\mathbf{x} - \mathbf{y}\| / \sigma$.

### A.3 DERIVATION OF CAUSAL RFA

This section presents a detailed derivation of causal RFA as in §3.1. Following Eq. 5 but changing the attended keys and values to the prefix:

$$\mathrm{RFA}(\mathbf{q}_t, \{\mathbf{k}_i\}_{i \leq t}, \{\mathbf{v}_i\}_{i \leq t}) = \frac{\phi(\mathbf{q}_t)^\top \sum_{i \leq t} \phi(\mathbf{k}_i) \otimes \mathbf{v}_i}{\phi(\mathbf{q}_t) \cdot \sum_{j \leq t} \phi(\mathbf{k}_j)} \tag{10}$$

Let $\mathbf{S}_t \triangleq \sum_{i \leq t} \phi(\mathbf{k}_i) \otimes \mathbf{v}_i$, and $\mathbf{z}_t \triangleq \sum_{i \leq t} \phi(\mathbf{k}_i)$; both can be calculated recurrently. Assuming $\mathbf{S}_0 = \mathbf{0}$ and $\mathbf{z}_0 = \mathbf{0}$:

$$\mathbf{S}_t = \mathbf{S}_{t-1} + \phi(\mathbf{k}_t) \otimes \mathbf{v}_t, \quad \mathbf{z}_t = \mathbf{z}_{t-1} + \phi(\mathbf{k}_t), \quad t \geq 1. \tag{11}$$

This completes the derivation of causal RFA as in §3.1.

### A.4 RFA WITHOUT NORM-1 CONSTRAINTS

§3.1 assumes that the queries and keys are unit vectors. This norm-1 constraint is *not* a must. Here we present a RFA *without* imposing this constraint. Let $C(\mathbf{x}) = \exp(\|\mathbf{x}\|^2 / 2\sigma^2)$. From Eq. 4 we have $\mathrm{attn}(\mathbf{q}_t, \{\mathbf{k}_i\}, \{\mathbf{v}_i\}) =$

$$
\begin{aligned}
\sum_i \frac{\exp(\mathbf{q}_t \cdot \mathbf{k}_i / \sigma^2)}{\sum_j \exp(\mathbf{q}_t \cdot \mathbf{k}_j / \sigma^2)} \mathbf{v}_i^\top &\approx \sum_i \frac{C(\mathbf{q}_t) C(\mathbf{k}_i) \phi(\mathbf{q}_t)^\top \phi(\mathbf{k}_i) \mathbf{v}_i^\top}{\sum_j C(\mathbf{q}_t) C(\mathbf{k}_j) \phi(\mathbf{q}_t) \cdot \phi(\mathbf{k}_j)} \\
&= \frac{\phi(\mathbf{q}_t)^\top \sum_i C(\mathbf{k}_i) \phi(\mathbf{k}_i) \otimes \mathbf{v}_i}{\phi(\mathbf{q}_t) \cdot \sum_j C(\mathbf{k}_j) \phi(\mathbf{k}_j)}.
\end{aligned}
\tag{12}
$$

The specific attention computation is similar to those in §3.1. In sum, lifting the norm-1 constraint brings an additional scalar term $C(\cdot)$.

### A.5 RELATING RFA-GATE TO SOFTMAX ATTENTION

Drawing inspiration from gated RNNs, §3.2 introduces a gated variant of RFA. Now we study its "softmax counterpart."

$$
\begin{aligned}
\widetilde{\mathbf{k}}_i &= \mathbf{k}_i (1 - g_i) \prod_{j=i+1}^t g_j, \quad \widetilde{\mathbf{v}}_i = \mathbf{v}_i (1 - g_i) \prod_{j=i+1}^t g_j, \quad i = 1, \dots, t \\
\mathbf{h}_t &= \mathrm{attn}(\mathbf{q}_t, \{\widetilde{\mathbf{k}}_i\}_{i \leq t}, \{\widetilde{\mathbf{v}}_i\}_{i \leq t}).
\end{aligned}
\tag{13}
$$

$\mathbf{h}_t$ is the output at timestep $t$ and is used for onward computation.

At each step, all prefix keys and values are decayed by a gate value before calculating the attention. This implies that the attention computation for $\mathbf{q}_{t+1}$ *cannot* start until that of $\mathbf{q}_t$ is finished. Combined with the linear complexity of softmax normalization, this amounts to quadratic time in sequence length, even for language modeling training.

The above model is less intuitive and more expensive in practice, without the RFA perspective. This shows that RFA brings some benefits in developing new attention models.

### A.6 DETAILED COMPLEXITY ANALYSIS

Table 4 considers a sequence-to-sequence model, and breaks down the comparisons to training (with teacher forcing; Williams & Zipser, 1989) and autoregressive decoding. Here we assume enough threads to fully parallelize softmax attention across timesteps when the inputs are revealed to the model in full. RFA has a lower space complexity, since it never explicitly populates the attention matrices. As for time, RFA trains in linear time, and so does the softmax attention: in teacher-forcing training a standard transformer decoder parallelizes the attention computation across time steps. The trend of the time comparison differs during decoding: when only one output token is produced at a time, RFA decodes linearly in the output length, while softmax attention decodes quadratically.

| Setting | Model | Time Complexity | | | Space Complexity | | |
|---|---|---|---|---|---|---|---|
| | | **Encoder** | **Cross** | **Causal** | **Encoder** | **Cross** | **Causal** |
| Training w/ teacher forcing | softmax | $\mathcal{O}(M)$ | $\mathcal{O}(M)$ | $\mathcal{O}(N)$ | $\mathcal{O}(M^2)$ | $\mathcal{O}(MN)$ | $\mathcal{O}(N^2)$ |
| | RFA | $\mathcal{O}(M)$ | $\mathcal{O}(M)$ | $\mathcal{O}(N)$ | $\mathcal{O}(M)$ | $\mathcal{O}(M+N)$ | $\mathcal{O}(N)$ |
| Decoding | softmax | $\mathcal{O}(M)$ | $\mathcal{O}(MN)$ | $\mathcal{O}(N^2)$ | $\mathcal{O}(M^2)$ | $\mathcal{O}(MN)$ | $\mathcal{O}(N^2)$ |
| | RFA | $\mathcal{O}(M)$ | $\mathcal{O}(M+N)$ | $\mathcal{O}(N)$ | $\mathcal{O}(M)$ | $\mathcal{O}(M+N)$ | $\mathcal{O}(N)$ |

Table 4: Time and space complexity comparisons between RFA and its softmax counterpart in a sequence-to-sequence attentive model, assuming an infinite amount of available threads. $M$ and $N$ denote the lengths of the source and target sequences respectively. Teacher forcing training (Williams & Zipser, 1989) and autoregressive decoding are assumed. Blue color indicates the cases where RFA asymptotically outperforms softmax attention.

| Data | Train | Dev. | Test | Vocab. |
|---|---|---|---|---|
| WikiText-103 | 103M | 218K | 246K | 268K |
| WMT14 EN-DE | 4.5M | 3K | 3K | 32K |
| WMT14 EN-FR | 4.5M | 3K | 3K | 32K |
| IWSLT14 DE-EN | 160K | 7K | 7K | 9K/7K |

Table 5: Some statistics for the datasets. WikiText-103 split sizes are in number of tokens, while others are in number of instances.

## B  EXPERIMENTAL DETAILS

Table 5 summarizes some statistics of the datasets used in our experiments. Our implementation is based on JAX.[15]

| # Random Matrices | 1 | 50 | 100 | 200 |
|---|---|---|---|---|
| **BLEU** | 24.0 | 25.7 | 25.8 | 25.8 |

Table 6: WMT14 EN-DE development set performance varying the number of random matrices to sample from during training. No beam search or checkpoint averaging is used.

During training, we sample a different random projection matrix for each attention head. Preliminary experiments suggest this performs better than using the same random projection throughout training (Table 6). Our conjecture is that this helps keep the attention heads from "over committing" to any particular random projection (Peng et al., 2020). To avoid the overhead of sampling from Gaussian during training, we do this in an offline manner. I.e., before training we construct a pool of random matrices (typically 200), at each training step we draw from the pool. At test time each attention head uses the same random projection, since no accuracy benefit is observed by using different ones for different test instances.

### B.1  LANGUAGE MODELING

We compare the models using two model size settings, summarized in Table 7. We use the fixed sinusoidal position embeddings by Vaswani et al. (2017). All models are trained for up to 150K gradient steps using the Adam optimizer (Kingma & Ba, 2015). No $\ell_2$-regularization is used. We apply early stopping based on development set perplexity. All models are trained using 16 TPU v3 accelerators, and tested using a single TPU v2 accelerator.

---

[15]https://github.com/google/jax.

| Hyperprams. | Small | Big |
|---|---|---|
| # Layers | 6 | 16 |
| # Heads | 8 | 16 |
| Embedding Size | 512 | 1024 |
| Head Size | 64 | 64 |
| FFN Size | 2048 | 4096 |
| Batch Size | 64 | 64 |
| Learning Rate | $[1 \times 10^{-4}, 2.5 \times 10^{-4}, 5 \times 10^{-4}]$ | |
| Warmup Steps | 6000 | 6000 |
| Gradient Clipping Norm | 0.25 | 0.25 |
| Dropout | [0.05, 0.1] | [0.2, 0.25, 0.3] |
| Random Feature Map Size | 64 | 64 |

Table 7: Hyperparameters used in the language modeling experiments.

## B.2 MACHINE TRANSLATION

**WMT14.** We use the fixed sinusoidal position embeddings by Vaswani et al. (2017). For both EN-DE and EN-FR experiments, we train the models using the Adam (with $\beta_1 = 0.1$, $\beta_2 = 0.98$, and $\epsilon = 10^{-9}$) optimizer for up to 350K gradient steps. We use a batch size of 1,024 instances for EN-DE, while 4,096 for the much larger EN-FR dataset. The learning rate follows that by Vaswani et al. (2017). Early stopping is applied based on development set BLEU. No $\ell_2$ regularization or gradient clipping is used. All models are trained using 16 TPU v3 accelerators, and tested using a single TPU v2 accelerator. Following standard practice, we average 10 most recent checkpoints at test time. We evaluate the models using SacreBLEU (Post, 2018).[16] A beam search with beam size 4 and length penalty 0.6 is used. Other hyperparameters are summarized in Table 8.

| Hyperprams. | WMT14 | IWSLT14 |
|---|---|---|
| # Layers | 6 | 6 |
| # Heads | 8 | 8 |
| Embedding Size | 512 | 512 |
| Head Size | 64 | 64 |
| FFN Size | 2048 | 2048 |
| Warmup Steps | 6000 | 4000 |
| Dropout | 0.1 | 0.3 |
| Cross Attention Feature Map | 128 | 128 |
| Causal Attention Feature Map | 64 | 64 |

Table 8: Hyperparameters used in the machine translation experiments.

## C MORE ANALYSIS RESULTS

### C.1 MORE RESULTS ON DECODING SPEED AND MEMORY OVERHEAD

Figure 3 compares the RFA's *unconditional* decoding speed and memory against the softmax attention. The setting is the same as that in §5 except that here the models do not have an encoder. This experiment aims to simulate the applications such as sampling from a language model.

### C.2 EFFECT OF RANDOM FEATURE SIZE

This section studies how the size of $\phi(\cdot)$ affects the performance. Table 9 summarize RFA-Gaussian's performance on WMT14 EN-DE development set. The model and training are the same as that used in §4.2 except random feature size. Recall from §2.2 that the size of $\phi(\cdot)$ is $2D$ for

---

[16]https://github.com/mjpost/sacrebleu

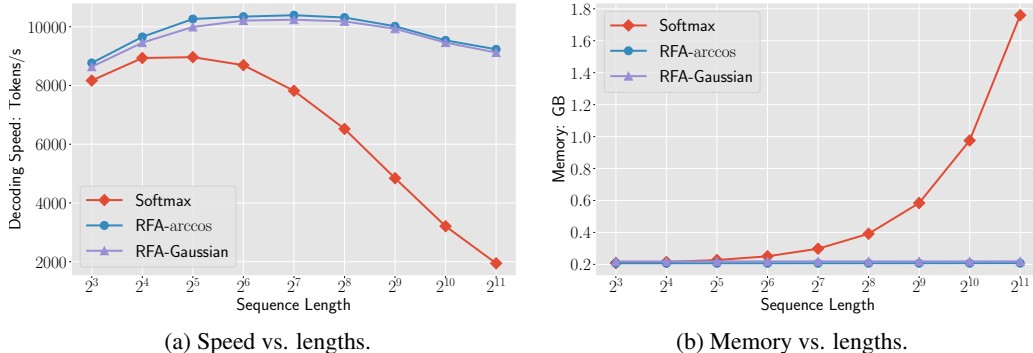

(a) Speed vs. lengths.  (b) Memory vs. lengths.

Figure 3: Unconditional decoding speed (left) and memory overhead (right) varying the output lengths. All models are tested on a single TPU v2 accelerator, with greedy decoding and batch size 16.

RFA-Gaussian. When the size of $\phi(\cdot)$ is too small (32 or 64 for cross attention, 32 for causal attention), training does not converge. We observe accuracy improvements by using random features sufficiently large (256 for cross attention and 128 for causal attention); going beyond that, the benefit is marginal.

| $\phi$ **Size** | 32 | 64 | 128 | 256 | 512 |
|---|---|---|---|---|---|
| **BLEU** | N/A | N/A | 24.9 | 25.8 | 26.0 |

| $\phi$ **Size** | 32 | 64 | 128 | 256 | 512 |
|---|---|---|---|---|---|
| **BLEU** | N/A | 25.3 | 25.8 | 25.8 | 25.6 |

(a) Varying cross attention $\phi$ sizes while fixing that of causal attention to be 128.

(b) Varying causal attention $\phi$ sizes while fixing that of cross attention to be 256.

Table 9: WMT14 EN-DE development set performance of RFA-Gaussian (the size of $\phi$ is $2D$; §2.2) varying the random feature sizes. N/A indicates training does not converge. No beam search or checkpoint averaging is used.

### C.3 TRAIN AND EVALUATE WITH DIFFERENT ATTENTION FUNCTIONS

RFA achieves comparable performance to its softmax counterpart. Does this imply that it learns a good approximation to the softmax attention? To answer this question, we consider:

(i) an RFA-Gaussian model initialized from a pretrained softmax-transformer;
(ii) a softmax-transformer initialized from a pretrained an RFA-Gaussian model.

If RFA's good performance can be attributed to learning a good approximation to softmax, both, *without* finetuning, should perform similarly to the pretrained models. However, this is *not* the case on IWSLT14 DE-EN. Both pretrained models achieve more than 35.2 development set BLEU. In contrast, (i) and (ii) respectively get 2.3 and 1.1 BLEU *without* finetuning, hardly beating a randomly-initialized untrained model. This result aligns with the observation by Choromanski et al. (2021), and suggests that it is *not* the case that RFA performs well because it learns to imitate softmax attention's outputs.

### C.4 KNOWLEDGE TRANSFER FROM SOFTMAX ATTENTION TO RFA

We first supplement the observation in Appendix C.3 by finetuning (i) on the same pretraining data. Figure 4 plots the learning curves. It takes RFA roughly 1,500 steps to reach similar training loss to the pretrained model. As a baseline, "RFA Reset" resets the multihead attention parameters (i.e., those for query, key, value, and output projections) to randomly initialized ones. Its learning curve is similar to that of (i), suggesting that the pretrained multihead attention parameters are no more useful to RFA than randomly initialized ones. To further confirm this observation, "softmax Reset"

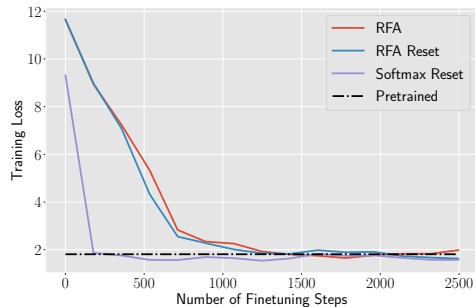

Figure 4: Finetuning an RFA-Gaussian model with its parameters initialized from a pretrained softmax-transformer. "Reset" indicates resetting the multihead attention parameters to randomly-initialized ones. The dashed line indicates the training loss of the pretrained model.

resets the multihead attention parameters *without* changing the attention functions. It converges to the pretraining loss in less than 200 steps.

**Takeaway.** From the above results on IWSLT14, pretrained knowledge in a softmax transformer *cannot* be directly transferred to an RFA model. However, from Figure 4 and a much larger-scale experiment by Choromanski et al. (2021), we do observe that RFA can recover the pretraining loss, and the computation cost of finetuning is much less than training a model from scratch. This suggests some potential applications. For example, one might be able to initialize an RFA language model from a softmax transformer pretrained on large-scale data (e.g., GPT-3; Brown et al., 2020), and finetune it at a low cost. The outcome would be an RFA model retaining most of the pretraining knowledge, but is much faster and more memory-friendly to sample from. We leave such exploration to future work.

