# OpenReview forum: "Random Feature Attention"
_ICLR.cc/2021/Conference — ICLR 2021 Spotlight_

### Official Review · AnonReviewer2 · 2020-10-28
**Interesting and impactful**

**Rating:** 8
**Confidence:** 4

**Review:**

##### Summary:
This paper presents Random Kernel Attention which is based on replacing the kernel function in the Linear Attention with random projection kernels. In general, I think the method is novel and quite impactful. Nowadays, some people are still staying away from attention because of its quadratic time and space complexity. To the best of my knowledge, it is the first attention method with linear complexity that can match or even outperforms the conventional attention.

##### Strengths:
- The method is intuitive and interesting to me.
- The results are strong. Unlike Linear Attention, the proposed RFA outperforms the original multi-head attention baseline on both LM and MT tasks. This is quite impressive. Based on my experience the Linear Attention with ELU non-linearity can bearly match the performance of the original attention mechanism.
- The authors provide several in-depth analyses in the appendix. I like the experiments in C.2.
- It is nice to see that the authors confess that the training is actually increased when using the RFA compared to the original Transformer. Usually, the inference time and memory usage are more important in practice.
- It is great to see that the authors compare with the baselines that cache the query/key/value representations. Nowadays, some papers avoid it make their speedup look better.


##### Weaknesses & suggestions:
- It is not clear what D is used in the experiments. The authors just vaguely say that they don't observe the improvement by setting it great than 2d. However, it would be better to see plots at least in the appendix. Also, I wonder if it would behave differently with different d. Also, it would be great to see what exactly the number is in the experimental setup to make this paper more reproducible.
- The arccos feature maps have only D-dimensional features, unlike the Gaussian feature maps which have 2D. It is not clear whether the authors use the same D for both variants or double the D of arccos to keep the feature dimensions the same.
- After introducing the random projection weights, the number of parameters would increase. It would be better if the number of parameters and the inference speed are both provided in Tables 1 & 2.
- The authors should clarify that the time complexity in Table 3 is based on the assumption that we have infinite number of threads or GPT/TPU cores that can be scaled up when M is increased. Otherwise, the time complexity of training the softmax model is still O(M^2) because there is a matrix multiplication between matrices of sizes M-by-M and M-by-d.

##### Questions:
- Based on the experiments, it seems that the Gaussian random feature maps don't really try to approximate
- How would the gating mechanism perform on the encoder side? Similar to BiLSTM, half of the dimensions can be applied in a backward manner to make it bidirectional.
- Do you resample the random weights during the time?
- I wonder if the authors will release their implementation. Based on my quick re-implementation, the proposed RFA doesn't really converge on some other dataset. I believe there might be some differences in how the parameters are initialized which is not clearly described in the paper. Admittedly, there is a chance that I have a bug in the code.
- Based on the conclusion in C.2 that RFA is not approximating the softmax kernel, would it be better if we just trained those projection matrices instead of fixing them as random matrices?

---

> ### Author Response · Authors · 2020-11-17
> **Response to Reviewer #2**
>
> Thanks for the detailed feedback! We are happy that R2 finds the paper intuitive and interesting!
> 1. About size of random feature $D$: as indicated in the “Setting” paragraphs of sections 4.1 and 4.2, in the language modeling experiments we use $D=64$, and in the machine translation experiment we use $D=64$ for causal attention and $D=128$ for cross attention. We will elaborate this detail. In preliminary experiments we explored how $D$ affects the performance. Please refer to bullet point 1 of the response to R3 for details.
> 2. R2 correctly pointed out that the arccos feature map is half the size of the Gaussian one, when the same $D$ is used. In the experiments we use the same $D$ for both feature maps, and this is the reason RFA-arccos slightly outperforms RFA-Gaussian in terms of time/memory efficiency (Figures 2 and 3).
> In early experiments we found that doubling the size of $D$ for arccos feature does not significantly improve its performance. We will clarify this in the revision.
> 3. The random projection weights are *not* trained, and do not add to the amount of parameters. In the tables all models have roughly the same amount of parameters. RFA only introduces a small amount of parameters: $\boldsymbol{\sigma}$ as in Eq. 9 ($d$ parameters per head), and those to implement the gate (if used, $d+1$ parameters per head). These accounts for less than 0.1% of the full model. We will clarify this.
> 4. Thanks for the great suggestion on the complexity analysis! This is definitely correct, which we missed when writing section 3.4. When the sequences are very long (e.g., >1000), it can happen that softmax attention can no longer be fully-parallized across timesteps even the full sequence is revealed to the model, due to the limited amount of threads. This suggests potential speedup from RFA in tasks such as text classification with long sequences, and this is exactly what we observe in the new experiment (Table 7). We will revise related discussion.
> 5. Thanks for the interesting suggestion on bidirectional RFA encoders! We will consider exploring it. Please refer to bullet point 5 of the response to R1 for some efficiency concern, which is the main reason we didn’t do this experiment.
> 6. Yes, we sample in an “offline” manner. Before training, we construct a pool of random matrices (typically 200). Each training step we sample, from the pool, a random matrix for each attention head. In this way the attention heads won’t “over commit” to a random matrix, and the overhead of sampling from Gaussian can be avoided. At test time each attention head uses the same random projection, since no accuracy benefit is observed by using different ones for different test instances. We will clarify this in the revision.
> 7. We will release our RFA implementation after the anonymity period.
> 8. On RFA not approximating softmax. We agree with R2’s interpretation of C.2. We conjecture that since we use relatively small $D$ for better efficiency, the approximation to softmax is less reliable. Then the approximation error is large enough that the model needs to learn to “accomodate” for it, resulting in a different training dynamic from softmax attention. This provides evidence that a normalization other than softmax could also be useful in attention.
> Training the projection matrices is an interesting direction! In preliminary experiments we explored a related (but different) idea: associate each head with a single projection matrix, which is learned from scratch along with the rest parameters of the model. In the machine translation experiments, it underperforms RFA and the softmax baseline by about 1 BLEU. We thank R2 for the great suggestion, which is definitely worth exploring!

---

> > ### Comment · AnonReviewer2 · 2020-11-20
> > **Response to authors' feedback**
> >
> > Thanks to the authors for answering my questions.
> > - Regarding the 1st point, I looking to seeing this ablation study in a revised version.
> > - Regarding the 6th point, it would be better to clarify this in the paper. I would encourage the authors to add some ablation studies on the pool size of random matrices to the paper.
> > - While I understand that the [performer paper](https://openreview.net/forum?id=Ua6zuk0WRH&noteId=UfTkeNE_oeN) is a concurrent work, it would be good to have some clarification of the differences between these two methods in the paper given they are quite similar.
> > - While I don't find the writing confusing to me, the criticisms by R4 about the writing look fair to me. I suggest the authors revise the confusing part during the rebuttal period.

---

> > > ### Author Response · Authors · 2020-11-23
> > > **Thanks for the followup!**
> > >
> > > Thanks for the suggestions! We will revise the paper accordingly.

---

### Official Review · AnonReviewer3 · 2020-10-28
**Yet another "linear attention", still, a good paper!**

**Rating:** 8
**Confidence:** 3

**Review:**

The paper presents a linear time and space attention mechanism based on random features to approximate the softmax. The paper is clearly written and easy to follow. The results are convincing: not chasing SOTA, but comparing to sensible baselines, namely [Baevski & Auli 2019] for language modeling on Wikitext-103, and [Vaswani et al. 2017] for machine translation on WMT14 EN-DE/EN-FR and IWSLT14 DE-EN.

The difference between theoretical speed-up and experimental speed-up is honestly discussed, and the overhead of the random features is not swept under the rug. However, having an experimental study on the random features dimensions' impact on empirical compute time vs. approximation performance vs. end-task performance would have been a plus.

I had read "Rethinking Attention with Performers" when I reviewed this paper, and I originally thought it was the same paper, but (along with notation being different) they start to differ on page 3. Where "Performers" goes with positive orthogonal random features (to improve over vanilla RFA), this paper adds a gating mechanism: this adds the possibility to learn some monotically decaying attention over older context, similar to learned receptive fields of attention (as in e.g. [Sukhbaatar et al. 2019]).

Overall, this is a good paper, and I don't see why we should downplay it in light of simultaneous ("Performers" got on ArXiV on September 30th, the ICLR deadline was October 2nd) quite similar contribution that the authors took the time to discuss. (It would be even better if they could compare to it in a future version.)

---

> ### Author Response · Authors · 2020-11-17
> **Response to Reviewer #3**
>
> Thanks for the insightful comments! We are glad that R3 finds our paper clearly-presented!
> 1. On random feature size vs. performance. In preliminary experiments we explored the effect of random feature size $D$ on the WMT14 EN-DE translation dataset. Here the attention heads are 64-dimensional. In decoder causal attention, when $D<64$ (e.g., 32, 16), the training is less stable and may not converge. With $D>64$ (e.g., 128, 256) the model is slower and consumes more memory, but does not improve accuracy much compared to $D=64$, which we use. There is a similar trend for cross attention, except that here we found $D=128$ strikes the best balance between accuracy and overhead. We will include this experiment in the revision.

---

### Official Review · AnonReviewer4 · 2020-10-30
**Interesting direction but the detailed/clarified discussions are needed**

**Rating:** 4
**Confidence:** 5

**Review:**

1. The logic in the introduction is a bit contradictive to me:

Some are able to achieve better asymptotic complexity (citations). while it
is more challenging to improve on shorter sequences: the additional computation steps required by
some approaches can overshadow the time and memory they save.

Doesn't this simply mean that for short sequences there is no such computational burden?
I think the story starts with pointing out the importance for long-sequence but turns to the topic on short sequence
which is confusing. The need for short sequence acceleration needs to be justified IMO.

2. Following 1, the baseline should be added.

For a fair comparison, I think the baseline should add those methods as claimed in the introduction
(Lee et al., 2019; Child et al., 2019; Sukhbaatar et al., 2019; Beltagy et al., 2020, inter alia), (Kitaev et al., 2020; Wang et al.,
2020; Roy et al., 2020, inter alia) and let us know how badly they performed under the short sequence. In particular we don't know
if the sacrifice of short sequence time would benefit a lot in long sequences for existing methods. The current experimental baseline can't reflect this.

3. Is the speedup over total computational time or just the attention part?

To best of my knowledge, under many circumstances in particular for short sequence, attention alone might not be the
most time-consuming part of the model. I think it will be helpful for authors to have a complete graph of the computational model used instead of only figure 1 concept graph. Specifically, is there
any feed-forward computation involved and how many layers of the models used in comparison.


4. Introduction of Eq 6,7 is confusing.

Up to eq 5 it's clear whatr's going, but it comes from nowhere to intorduce these 2 modules in eq 6,7. So my understanding is that
RFA simply refers to the approximation of computing the softmax. So the statement:

for softmax-attention. The latter is typically used in two different ways in the transformer architecture, each resulting in a different computation for RF

is confusing as the RFA is now redefined. I believe RFA should only refer one thing and I don't think eq(6) and eq(5) leads to the
same result. On the other hand, eq 7 should be the same as eq 5. Is this correct?
In addition, the notation in (6) looks wrong to me. \phi(x) as introduced in eq 2 is in R^{2D} but S_{t-1} is in R{D}, not sure what does + mean in this context.
I couldn't find out where you properly define the meaning of D either.


5. Clarification of contribution

Eq 6,7 reads like RNN style update but the intuition is lacking. Do you want to claim that this structure design is inspired by RNN and it leads to a better result?
Put in another way, using RFA in transformer is from Rawat et al., 19 so do you think your major contribution is to design such
a gated usage of RFA?


5. Discussion of D

Since RF is not the major contribution, you summarize existing results of FA in sec2.2. I think I'd like to see a discussion of
sufficient number D analytically or empirically. Could you also cite the convergence bound on this approximation? To me, D looks to be an important efficiency tradeoff. Say sequence length is M
and feature in d dimension. Original Attention is O(M^2 d). The computation of RFA
requires outer product, which is O(D^2d) so overall it's O(M D^2 d), if M is around 64 or 128 (common usage) and D is 64, I actually
don't see why RFA could improve 2x. Do you pre-compute and pre-store anything?


6. Time analysis on language modeling is not presented. Since it's a efficiency paper, I think it should be complete.



Overall, I think the paper provides an interesting view of discussion, but there are many flaws in the current version which needs to be corrected before a more serious consideration.  Especially, in terms novelty, the paper is relatively limited as the RF is explored in Rawat et al., 19. So my point 5 is important to answer and I would like to see all the details are clarified in order to make the contribution stronger.

---

> ### Author Response · Authors · 2020-11-17
> **Response to Reviewer #4, 1/2**
>
> Thanks for the insightful review!
> - Clarification of contribution: the contribution of this work is three-fold. (a) Using random feature methods in attention and transformers. Rawat et al. (2019) applied random feature methods to speed up softmax calculation over large vocabularies, but they didn’t explore its use in attention or transformers. (b) Inspired by the causal RFA (Eq.6)’s connection to recurrent neural nets, we propose a gated variant of RFA which helps promote recency when such a learning bias is desired. (c) RFA achieves strong performance on real-world NLP tasks including both MT and LM. To the best of our knowledge, RFA is the first efficient transformer variant to achieve significant decoding speedup on MT without accuracy loss. Further, augmented with a gating mechanism, RFA outperforms the strong transformer baseline on WikiText-103 LM dataset in terms of perplexity.
> 1. Thanks for pointing out that our wording may cause some confusion. Transformers are central to many NLP tasks, and it is important to make them more efficient at inference time. While exciting progress has been achieved by existing approaches that focus on long sequences, our goal is to develop a model that improves efficiency for both long and moderate length sequences, which opens up the possibility to apply our approach in many real-world NLP settings. We will elaborate this in the revision.
> 2. Thanks for raising this point! Indeed, many recent works have explored improving transformers’ efficiency. We do compare to Katharopoulos et al. [10], a very recent effort, in both experiments, and show that RFA performs better in terms of accuracy. In an additional experiment, we use the setting by Tay et al. [2] and compare RFA against some of the mentioned related works including Child et al. [3], Beltagy et al. [4], Wang et al. [5], Kitaev et al. [6], Tay et al. [7], Tay et al. [8], Zaheer et al. [9], Katharopoulos et al. [10], and Chromomanski et al. [11]. In terms of both accuracy and time/memory efficiency, RFA is among the strongest. Preliminary results are presented in Table 7. In the revision, we will finalize this experiment and include it in the main text.
> 3. The speedup is over total computation time. Thanks for raising the point that feed-forward components can be another time-consuming part. This is definitely correct, and makes it more challenging to achieve wall-clock time speedup. All of the speed comparisons are made between full models, including both attention components and modules such as FFN, layer normalization, output softmax over the vocabulary. The machine translation experiments use 6-layer encoders and decoders, and so does Figure 2 (a); Figure 2 (b) compares 6-layer decoders. Thanks for the suggestion on the diagram! We will clarify these details in the revision.
> 4. Thanks for pointing out the typo in Eq.6, which we have fixed. Both $\mathbf{S}_t$ and $\mathbf{S}_\{t-1\}$ should be of $\mathbb{R}^\{2D \times d\}$ here.
> Both Eq.5 and Eq.6 describe computation of RFA. Their differences rise from the context they attend to: Eq.5 attends to the full sequence, while Eq. 6 attends only to the prefix. We’ve revised this part.
> Here + denotes elementwise summation between matrices or vectors, and $2D$ is the size of $\boldsymbol{\phi}(\boldsymbol{\cdot})$. It is correct that Eq.7 is the same as Eq.5. We have updated the notations, and removed the old Eq.7 for better consistency.

---

> > ### Comment · AnonReviewer4 · 2020-11-21
> > **Clarification on 2 points.**
> >
> > On this part of the response, I have two questions.
> >
> > 1) I still couldn't get the main contribution of the work. In my view, the random feature is the key to accelerate but it's not new. So it's important to demonstrate the capability of Causal RFA, but honestly I still can't get it clear how the derivation steps of the Causal part after reading again the paper. It will be great if you could update the draft to make it clear.
> >
> > In addition, it seems like Random Feature is also explored by performer https://arxiv.org/abs/2009.14794 . I feel it's almost the same as the proposed usage in the paper as it's again using random feature to accelerate softmax thus transformer. I don't see the major difference. Thus I don't think RF is a contribution. The only contribution I acknowledge is Causal RFA part, which is rather limited. I personally think this is totally acceptable in research progress nowadays; however, the experimental and writeup part(at least I am still unclear) needs to be fairly clear and it leads to my point 2.
> >
> > 2) I am extremely confused by wall-clock time claim. I think you need to reveal more on the models you timed. For example, if it's a normal transformer-base model used in some popular framework like huggingface, the attention dimension is 768 but there are 2 or 3 layers of feed-forward layer which has a large size 768 x 3072. So take context size to be 128, I believe by counting the computation needed in the model it's impossible to achieve 2x+ speedup as shown in the paper. The attention is like a 128 x 768 x 768 matrix transformation computation plus  self-attention is 128 x 128 x 768 computation. But the FFN is 128 * 768 * 3072. In other word, I think under the basic setup the FFN will dominate the computational time according to the model size setup. It's impossible to achieve a 2x speedup if attention module is not the dominating computational module (because if FFN dominates, you need to at least make attention 0 computational time to have 2x). I could understand it will become so if context size becomes larger but the 2x speedup is reported in the small N setup which really confuses me. Could you reveal how you do the profiling? (including hardware, model size, profiling methods, computational time of each module). To me it doesn't make sense from counting the model parameters.

---

> > > ### Author Response · Authors · 2020-11-23
> > > **Thanks for the followup!**
> > >
> > > 1. We kindly remind R4 that even though random feature methods are well-established, their application in attention and transformer is new. Besides, as we emphasized in previous response, we propose a gated variant of RFA which helps promote recency learning bias and proves useful in our language modeling experiments. Furthermore, to the best of our knowledge, RFA is the first efficient transformer variant that achieves substantial decoding speedup on machine translation without accuracy loss.
> > > The performer paper R4 pointed to is a concurrent work under review at the same ICLR 2021 conference, as R2 and R3 correctly pointed out. Although it is not required, we are curious to know how RFA compares to Performer in NLP tasks. We conducted additional experiments with three text classification datasets. The results show that RFA consistently outperforms Performer in terms of accuracy, with comparable time/memory efficiency. We’ve included some of the results into the paper, and will add the rest in the revision.
> > > We will present a step-by-step derivation of causal RFA.
> > > 2. Thank you for the comment. We would like to clarify the walltime computation.
> > > - First, as emphasized throughout the paper, the 2x speedup is achieved in sequence-to-sequence machine translation models at decoding time. The setting is very different from the huggingface example raised by R4. In decoding, the computation at timestep $t$ *cannot* start until that of timestep $t-1$ is finished. Further, as per standard practice, the decoder of a transformer seq2seq model first attends to the prefix, then attends to the encoder outputs, and lastly feeds the outputs to an FFN. In other words, each decoding step performs two attention steps, and a single FFN step. For details please refer to a well-established implementation of the transformer decoder by fairseq: https://github.com/pytorch/fairseq/blob/master/fairseq/modules/transformer_layer.py#L344-L398
> > > - Second, R4 correctly points out that in transformers, FNN hidden layer size is usually much larger than sequence length. However, this does *not* imply that running a 3,072-sized FNN is 24 times slower than attending over a 128-length sequence. One of the main reasons for this is that the large matrix multiplication in the FFN is much more friendly to modern accelerators than softmax attention is.
> > > - Third, our speed is measured in tokens per second. The models greedily decode multiple outputs, which is timed with the `time` module in Python. We believe this reflects real-world applications well, but are open to suggestions. The models are of the same size as those used in the machine translation experiments, with 6 encoder/decoder layers, 8 attention heads, 512 embedding size, 64 head size, and 2048 FFN hidden layer size (Table 6). The decoding speed of all models is tested on a single TPU v2 accelerator, as indicated in Table 2 and Figure 2.
> > > - To address R4’s concern that it is the FFN that dominates the computation time, we’ve included in the supplementary material an anonymous code snippet aiming to compare the computation time of softmax attention against FFN’s in a sequence-to-sequence decoding setting. It uses the same auto-diff toolkit as that used in our experiments to sketch our decoding implementation, but leaves out some less relevant modules (e.g., layer normalization, embedding, output softmax over vocabulary) for clarity. On a single TPU v2 accelerator (which we use in all speed and memory related experiments), the attention computation takes around 6.2x more time than the FFN. We note that the results by this snippet might show different trends from those in the paper, which instead compare full models.

---

> ### Author Response · Authors · 2020-11-17
> **Response to Reviewer #4, 2/2**
>
> 5. Discussion of $D$. It is correct that the choice of $D$ largely affects the efficiency, as discussed in section 3.4.
> We beg to differ regarding the overhead related to outer product. Taking the outer product between a $2D$-dimensional vector and a $d$-dimensional vector takes $\mathcal{O}(Dd)$ time and space, instead of $\mathcal{O}(D^2d)$. And since such computation is highly parallelizable on modern accelerators, in practice its time overhead is fairly small. For the full sequence case, RFA never explicitly constructs the $\mathcal{O}(MDd)$ tensor, but only stores $\mathbf{S}$ and $\mathbf{z}$ in the memory with $\mathcal{O}(Dd + d)$ space, and calculates the outer product for each token one at a time as it reads through the sequence. So the overall space complexity is $\mathcal{O}(Dd)$ instead of $\mathcal{O}(MD^2d)$.
> It is a good point that pre-computation can help achieve saving, which is what we do. In the machine translation experiments: the encoder outputs are stored into $(\mathbf{S}, \mathbf{z})$ tuples for the cross attention; each decoding step the query (after random feature map) multiplies against $\mathbf{S}$ and $\mathbf{z}$, instead of explicitly attending to the M-length source. It is similar for the decoder self attention. Since matrix multiplication is very friendly to modern accelerators, 2x speedup can be achieved even when $D$ is similar to $M$. We will include a detailed discussion on this.
> The convergence bound of random features methods is well-studied. We will discuss it citing related results.
> 6. We did not achieve any speedup in the language modeling experiments, as indicated in Section 3.4 and footnote 12. This is because in our LM experiment, the softmax attention computation can be parallelized across timesteps. As a result its time complexity is already linear, so RFA does not have any advantage at training time or evaluating perplexity. However, with a simulation experiment, we do find that RFA achieves significant speedup and memory saving when sampling from a language model (Figure 2b). We will elaborate these empirical results.
>
> [1] Ankit Singh Rawat, Jiecao Chen, Felix Xinnan X Yu, Ananda Theertha Suresh, and Sanjiv Kumar. Sampled softmax with random fourier features. 2019. In Proc. of NeurIPS.
>
> [2] Yi Tay, Mostafa Dehghani, Samira Abnar, Yikang Shen, Dara Bahri, Philip Pham, Jinfeng Rao, Liu Yang, Sebastian Ruder, Donald Metzler. Long Range Arena: A Benchmark for Efficient Transformers. 2020. arXiv:2011.04006.
>
> [3] Rewon Child, Scott Gray, Alec Radford, and Ilya Sutskever. Generating long sequences with sparse transformers, 2019. arXiv:1904.10509.
>
> [4] Iz Beltagy, Matthew E. Peters, and Arman Cohan. Longformer: The long-document transformer, 2020. arXiv:2004.05150.
>
> [5] Sinong Wang, Belinda Li, Madian Khabsa, Han Fang, and Hao Ma. Linformer: Self-attention with linear complexity. 2020. arXiv:2006.04768.
>
> [6] Nikita Kitaev, Lukasz Kaiser, and Anselm Levskaya. Reformer: The efficient transformer. 2020. In Proc. of ICLR.
>
> [7] Yi Tay, Dara Bahri, Donald Metzler, Da-Cheng Juan, Zhe Zhao, and Che Zheng. Synthesizer: Rethinking self-attention in transformer models. 2020. arXiv:2005.00743.
>
> [8] Yi Tay, Dara Bahri, Liu Yang, Donald Metzler, and Da-Cheng Juan. Sparse sinkhorn attention. 2020. arXiv:2002.11296.
>
> [9] Manzil Zaheer, Guru Guruganesh, Avinava Dubey, Joshua Ainslie, Chris Alberti, Santiago Ontanon, Philip Pham, Anirudh Ravula, Qifan Wang, Li Yang, et al. Big bird: Transformers for longer sequences. 2020. arXiv:2007.14062.
>
> [10] Angelos Katharopoulos, Apoorv Vyas, Nikolaos Pappas, and Francois Fleuret. Transformers are rnns: Fast autoregressive transformers with linear attention. 2020. In Proc. of ICML.
>
> [11] Krzysztof Choromanski, Valerii Likhosherstov, David Dohan, Xingyou Song, Jared Davis, Tamas Sarlos, David Belanger, Lucy Colwell, and Adrian Weller. Masked language modeling for proteins via linearly scalable long-context transformers. 2020. arXiv:2006.03555.

---

### Official Review · AnonReviewer1 · 2020-11-02
**Linear-time attention with a gating mechanism that is both more accurate and faster than standard softmax attention**

**Rating:** 8
**Confidence:** 3

**Review:**

Summary
* This paper proposes a linear time and space attention variant that matches (or exceeds) the accuracy of standard attention while maintaining the speedup of prior work in linear time/space attention.
* The approach is centered around a linear approximation of softmax attention, and is extended with a gating mechanism similar to a GRU.
* The accuracy improvements from the gated extension are demonstrated via language modeling on WikiText-103, while the speed-ups are demonstrated on translation.

Contributions
* Demonstrates the importance of the choice of kernel in RFA, as prior results from Katharopoulos et. al [1] underperforms softmax attention, while this work gets comparable performance by changing the kernel.
* Proposes an extension of RFA with gating that improves accuracy on language modeling, relative to softmax attention.

Strengths
* The RFA-gated formulation is both more accurate and potentially faster (at least for decoding) than softmax attention, as demonstrated on language modeling.
* The writing was clear and thorough.
* The experiments support the claims of improved accuracy via choice of kernel and gating, and preservation of speedups inherited from the linear attention formulation.

Weaknesses
*  The improved accuracy and speed over the baseline transformer are great, and the experiments serve to nicely illustrate those independently. I would like to see a third application where these two qualities are demonstrated together.
* The gating mechanism could also be applied to the output of softmax attention, but that comparison is not included. Please correct me if this is incorrect.

Recommendation: Weak Accept
* Well-written and timely exploration of linear attention.
* Empirical results demonstrate accuracy improvement over softmax attention, while preserving the linear time complexity.
* Extension of RFA with a gating mechanism appears to be effective, but I do not believe the claim that it is hard to apply this to softmax attention is valid, leaving the main contribution an exploration of a couple different kernels for linear attention.

Questions
* Can you clarify how the RFA defined in section 3.1 differs from the linear attention in Katharopoulos et. al. [1]? What is different other than the choice of phi?
* Associative reductions such as those described in sections 3.1 and 3.2 can be computed in time logarithmic in sequence length (at the cost of n log n memory consumption) on a parallel device using a binary reduction or the prefix sum trick. Would this result in speed gains, or is the performance of attention in the parallelizable softmax setting already saturated?
* Are there experiments with a bidirectional gated RFA in the conditional setting, i.e. in the encoder for translation?

Nits
* The formatting of equation 5 is a bit strange, as the start of the equation is inline while the rest is not.

[1] Angelos Katharopoulos, Apoorv Vyas, Nikolaos Pappas, and Francois Fleuret. Transformers are rnns: Fast autoregressive transformers with linear attention. In Proc. of ICML, 2020.

Edit: I have updated my rating based on author response.

---

> ### Author Response · Authors · 2020-11-17
> **Response to Reviewer #1**
>
> We thank R1 for the valuable comments!
> 1. Additional experiments. We additionally tested out RFA as encoders, on a character-level text classification task. It is a binary sentiment classification task with the IMDb movie review data. Following the same setting as Tay et al. [1], RFA is among the strongest systems in terms of both accuracy and efficiency. Preliminary results are summarized in Table 7. In the revision we will finalize this experiment and include it in the main text.
> 2. Applying gating to the output of softmax attention: R1 raised a great point that it is possible to promote recency learning bias by applying gating at the output of the softmax attention. As we will detail later, this does not directly relate to RFA-Gate, in the sense that it is not the model that RFA-Gate approximates.  Assuming the causal attention, this corresponds to the following model:
> \begin{align}
> \mathbf{z}_t&=\operatorname{attn}(\mathbf{q}_t, \\{\mathbf{k}_i\\}_\{i\leq t\}, \\{\mathbf{v}_i\\}_\{i\leq t\})\\\\
> \mathbf{y}_t &= g_t\mathbf{y}_\{t-1\} + (1-g_t)\mathbf{z}_t.
> \end{align}
> $\mathbf{z}_t$ denotes the attention output at timestep $t$, and $\mathbf{y}_t$ is the output after the gating mechanism, which is then fed into the output affine transformation. Scalar sigmoid gate $g_t$ is implemented in the same way as that in RFA-gate. Note that the attention here is the same as the softmax attention, which does not explicitly include recency bias. We tested out this model on WikiText-103 with the small-sized setting, it achieves a 35.0 test perplexity, slightly worse than the softmax attention baseline.
> There is indeed a “softmax counterpart” of RFA-Gate-Gaussian,
> \begin{align}
> \widetilde{\mathbf{k}}_i &= \mathbf{k}_i \  (1-g_i) \prod_\{j=i+1\}^\{t\}g_j,
> \quad \widetilde{\mathbf{v}}_i = \mathbf{v}_i  \ (1-g_i) \prod_\{j=i+1\}^\{t\}g_j, \quad i=1,\dots,t \\\\
> \mathbf{h}_t&=\operatorname{attn}(\mathbf{q}_t, \\{\widetilde{\mathbf{k}}_i\\}_\{i\leq t\}, \\{\widetilde{\mathbf{v}}_i\\}_\{i\leq t\}).
> \end{align}
> At each step, all prefix keys and values are decayed by a gate value before calculating the attention. This implies that the attention computation for $\mathbf{q}_\{t+1\}$ cannot start until that of $\mathbf{q}_t$ is finished. Combined with the linear complexity of softmax normalization, this amounts to quadratic time in sequence length, even for language modeling training. In contrast, both RFA-Gate and the canonical softmax attention are of linear time complexity at training time. We are not able to finish training it on WikiText-103 at the time of this response. We will try to include the results in the revision if time permits.
> The above $\mathbf{h}_t$ model is less straightforward to see without the RFA perspective, and is practically more expensive without RFA’s formulation. This shows that RFA brings some benefits in developing new attention models. We will include this discussion in the revision.
> 3. When used in causal attention, RFA without gating is related to Katharopoulos et al. [2]. The key difference is that this previous work uses a $\mathrm{elu}(\cdot)+1$ feature map; RFA uses a well-established random feature map to approximate softmax attention, which achieves strong performance in our experiments. We also derive its encoder self attention and across attention counterparts.
> 4. R1 raised a great point that summing a sequence of elements could be done in logarithmic time in the number of elements, using a divide and conquer strategy. To the best of our knowledge, this is not implemented by modern auto-diff toolkits or backend libraries such as CUDA kernels. Our conjecture is that as the summation proceeds, less and less threads are needed, which leads to an inefficient way of single-program-multiple-data parallelization. Further, this algorithm could involve much more read/write operations, which can be the key bottleneck on modern accelerators such as GPUs. Last but not least, it is much more difficult to implement. We leave this to future work but are not optimistic.
> 5. Thanks for the great suggestion! One benefit of gated RFA is the bidirectional encoder, reminiscent of Bi-RNN encoders, which is definitely worth exploring. We did not include such an experiment, mainly because for MT sequences (usually less than 100 tokens), RFA does not yield any speed gain when applied to encoders, as discussed in 3.4 (in this case, a bidirectional RFA may actually double the encoding time). We will consider including this experiment for the revision if it can be carried out on this project’s budget.
> 6. We've fixed Eq. 5.
>
> [1] Yi Tay, Mostafa Dehghani, Samira Abnar, Yikang Shen, Dara Bahri, Philip Pham, Jinfeng Rao, Liu Yang, Sebastian Ruder, Donald Metzler. Long Range Arena: A Benchmark for Efficient Transformers. arXiv:2011.04006
>
> [2] Angelos Katharopoulos, Apoorv Vyas, Nikolaos Pappas, and Francois Fleuret. Transformers are rnns: Fast autoregressive transformers with linear attention. In Proc. of ICML, 2020.

---

### Author Response · Authors · 2020-11-24
**Summary of paper revision**

We thank AC for handling this paper and the reviewers for their insightful feedback! We have thoroughly revised the paper according to their suggestions and comments. Major changes include:
1. We have included additional experiments on long text classification aiming to compare our model against recently proposed  approaches when used as text encoders (Section 4.3). It achieves strong accuracy on all three datasets, and is among the most competitive in terms of efficiency.
2. We’ve included ablation studies on the number of random matrices (Appendix B), random projection size (Appendix C.2).
3. We present a step-by-step derivation of causal RFA in Appendix A.3, and discuss RFA-Gate’s relation to softmax attention in Appendix A.5.
4. We’ve addressed most of the reviewers’ comments.

The revisions are highlighted in red; in the next version we will remove the highlighting.

Thanks,

Paper1925 Authors

---

### Decision · Program_Chairs · 2021-01-07
**Final Decision**

**Decision:**

Accept (Spotlight)

**Comment:**


This paper proposes an efficient attention mechanism linear in time and space using random features.
The approach has some similarities with the simultaneous ICLR 2021
submission "Rethinking Attention with Performers", with a key difference of a gating
mechanism present in this work, motivated by recency bias. This paper is a
valuable contributions to the efficient attention research topic. The reviewers
appreciate the experiments and the in-depth analysis. I recommend acceptance.

A noteworthy concern brought up in the discussion period has to do with whether the attention mechanism dominates the feed-forward computations in the neural network, and how much this is architecture-specific. The authors provide TPU timings, but I encourage the authors to add a discussion and timings of relative performance of feed-forward vs. attention layers that covers GPU and CPU optimizers as well.